# Instrumented Nanoindentation Tests Applied to Bulk Metallic Materials: From Calibration Issue to Pile-Up Phenomena

**DOI:** 10.3390/ma14216360

**Published:** 2021-10-24

**Authors:** Marcello Cabibbo

**Affiliations:** Dipartimento di Ingegneria Industriale e Scienze Matematiche (DIISM), Università Politecnica Delle Marche, Via Brecce Bianche 12, 60131 Ancona, Italy; m.cabibbo@staff.univpm.it; Tel.: +39-071-220-4728

**Keywords:** nanoindentation, calibration, pile-up, metallic materials

## Abstract

Instrumented nanoindentation tests have reached an effective level of theoretical and practical knowledge to become an interesting and useful tool for determining hardness, *H*, and local elasticity (reduced Young’s modulus), *E_r_*, of a variety of materials, from coatings and thin films to bulk metallic materials. Nanoindentation instruments are equipped with analysis software for raw data for hardness and reduced Young’s modulus evaluation, generally based on the Oliver and Pharr analysis method. On the other hand, it is widely known and recognized that prior data acquisition, a tip-dependent calibration procedure of compliance, and area function are needed. With this in view, an accurate and sound calibration protocol is here reported. Hardness and local elastic modulus is measured on different bulk metallic materials, showing the distinctive strengths of using nanoindentation. Finally, a local elastic-plastic phenomenon mostly induced by the nanoindentation tip on ductile metallic material (i.e., pile-up) is also reported and modelled. This manuscript is thus intended to favor and account for the importance of using the instrumented nanoindentation tests for *H* and *E_r_* measurements of metallic materials.

## 1. Introduction

Nanoindentation is a relatively new technique for the measurement of hardness, *H*, reduced Young’s modulus, *E_r_*, and strain rate sensitivity when small penetration depths and measurement volume are required [1,2,3,4,5,6]. Since the instrument measures these mechanical data through a coupling procedure between the indentation tip and the testing material, and then though post-measurement load-displacement analyses, the system calibration and analysis procedures are of primary importance. The usefulness of nanoindentation, among different metallurgical aspects, lies in its ability to evaluate hardness and local elastic modulus by means of a single set of measurements. Hence, nanoindentation can be considered a straightforward material yield strength evaluation method that has the advantage of not requiring preparation of specific sample geometry [1,2,3].

The nanoindentation hardness is measured by evaluating the contact area between the two materials that have come into contact: the hard tip and the testing material. This can be affected by the sample surface roughness; by some existing microstructure features, such as grain structure, interfaces, precipitates, and so forth; and by the ductile response of the sample [1,3].

Nanoindentation tips can be in form of a sharp triangular pyramid (such the Berkovich), cube corner indenters, or simply in the form of spheres. The former two are generally used when the smallest possible length scale is needed; the latter is generally used for acquiring information on the elastic behavior of coatings and thin films [1,4,5,7]. This is because the process of indentation contact on the sample surface tends to be simply elastic when the load is low, while it turns into elastic-plastic as the load rises, from which point the evaluation of the sample yield strength is possible [4,5]. Sharp tips and spherical indenters suffer from sharpness imperfections or exact spherical geometry, respectively. These generally appear after several thousands of hours of test. It is then straightforward that this geometric deterioration can lead to measurement inaccuracies, thus affecting the nanoindentation measurements.

The indenter penetration depths can be as low as 20 nm with very limited contact areas. Thus, the accuracy of the tip radii and spherical geometry is of the utmost importance [4,5]. On this ground, the tip calibration procedures are quite important to assess the soundness of the mechanical measurements obtained by nanoindentation test. A further aspect is the instrument-sensitive calibration set-up procedures. In fact, to obtain very accurate *H* and *E_r_* measurements starting from the recorded raw data, each nanoindentation instrument requires specific calibration procedures to be followed.

Despite the hardware variations among different instruments, the Oliver and Pharr approach is widely recognized as the reference method for the raw data analysis. On this ground, the ISO/FDIS 14577-1 standard [8] complies with the Oliver and Pharr method [9] for both *H* and *E_r_* evaluation through nanoindentation. This standard is indeed an attempt to provide guidelines to guarantee the reproducibility and direct comparativeness of the nanoindentation results among the different existing instruments.

On the other hand, the advantage of using nanoindentation against Vickers micro-hardness tests, namely for metallic materials, lies in the unique possibility to directly correlate submicrometric (and sometimes even nanometric scale) microstructure features to the material mechanical characteristics of *H* and *E_r_*. Nanoindentation measurements of bulk metallic materials have the possibility to infer local mechanical properties of specific features, such as pearlite colonies in steels, single-phase analysis in multi-phase and multi-component metallic materials, and secondary phase particles in light alloys, such as aluminum, magnesium, and so forth [1,8,9].

In addition, ductile metallic materials are known to produce an elastic-plastic boundary deformation all around the produced tip print. This is called the pile-up phenomenon and the corresponding extension from the tip print edge outward strongly depends on the material ductility response [10,11,12,13]. This local, important coupling aspect between the tip and the material surface is likely to be used as a further tool by nanoindentation to determine the ductile response and behavior of bulk metallic materials, and eventually, to infer the mechanical properties of some of the sub-micrometer microstructure features.

This work deals with the optimization of calibration procedures, application of post-calibrated nanoindentation to different metallic materials, and use of the pile-up phenomenon in ductile pure copper for a ductility response evaluation by nanoindentation.

## 2. Set-Up, Materials, and Method

### 2.1. Calibration Set-Up and Protocols

Nanoindentation calibration is performed by using one of the commercially available reference samples, depending on the type of material to be tested: fused quartz (Fq: *E_r_* = 72 GPa, Poisson ratio, *ν* = 0.17) as hard probe, sapphire (Sa: *E_r_* = 410 GPa, *ν* = 0.234) as high-elastic-modulus probe, polycarbonate (Pc: *E_r_* = 3.3 GPa, *ν* = 0.37) as soft probe. The latter one, due to its low elastic modulus, is seldom used for *H* and *E_r_* measurements of metallic materials. The proposed calibration procedure (protocol) refers to the mostly used indenters: Berkovich (B), cube corner (cc), and spherical (Sp).

The measurement protocol consists of two steps. The first one refers to the stiffness (which for the Oliver and Pharr data analysis is the inverse of the compliance) and the area function calibration. The second one refers to the *H* and *E_r_* measurements performed by applying the first calibration step. The two-step protocol was extended to all three different indentation tips. As for the first calibration step, stiffness and area functions were calibrated on Fq, Sa, and Pc for all three indenters, with a trapezoid-shaped function consisting of a load time, *t_load_* = 10 s; a holding time to the maximum load, *t_hold_* = 5 s; and a final unloading time, *t_unload_* = 10 s. The selected maximum load, *P_max_*, was in the range of 100–10,000 µN. This set-up is schematically depicted in Figure 1. As for the second step of the calibration protocol, the *H* and *E_r_* measurements were calibrated on Fq and Sa using functional load, as reported in the schematic representation of Figure 2a; on the other hand, when using Pc, function loads had a two-stage character, as schematically represented in Figure 2b.

The *H* and *E_r_* values obtained by calibration on Fq were measured using a cyclic load function with t_load_ = 10 s, t_hold_ = 10 s, and t_unload_ = 3 s, and consisted of 10 steps of load progression by 10% up to the maximum machine load, and then unloading to a final holding time of 120 s at 10% of the maximum reached load. The maximum machine load was 10 mN (Hysitron^TM^ Triboscope UBI-1^®^). To have a statistically sound result, all measurement procedures of the calibration protocol were repeated 10 times.

The second step of the calibration protocol was carried out on the three different nanoindentation indenters—fused quarts (Fq), sapphire (Sa), and polycrystal (Pc)—and turned out to be the most accurate method of set-up and calibration procedure. This was due to the different stiffness and elastic modulus of the three mentioned indenters.

The latter calibration protocol was selected for the nanoindentation measurements on materials of practical interest.

### 2.2. Materials

The selected materials for testing ranged from pure single-phase metals (commercially pure aluminum and DHP copper) to alloyed steel (fully annealed 100 Cr6 ball bearing steel) and ferritic-pearlitic cast iron. The sample surfaces were mechanically polished to minimize mean roughness (within 30 nm) and obtain a perfectly smooth and planar surface. Conventional chemical etchings were then used to reveal the microstructures of the different materials. Namely, a solution consisting of 5% hydrofluoric acid and 95% distilled water was used for pure aluminum, while 2% Nital was used for DHP copper, 100 Cr6, and cast iron.

A Hysitron© Triboscope (UBI-1^®^) equipped with a Berkovich indenter ending with a tip of apex radius of 70 ± 5 nm was employed.

The soundness of nanoindentation hardness measurements were inferred by comparing them to microhardness tests; the latter were carried out using a Vickers Remet© HX-1000^®^ with a load of 200 g and dwelling time of 15 s. Microhardness tests were performed according to the UNI EN ISO 6507-1. A Leika DMi8 was used to carry out light microscopy.

### 2.3. Pile-Up in Ductile Metallic Materials

It is indeed known that in ductile bulk metals, the elastic-plastic loading response upon tip contact is likely to generate material pile-up around the indenter, and then around the tip print afterwards [14,15,16].

In this regard, according to Tabor [14], for a sharp conical or pyramidal indenter, the measured hardness must be independent of the applied load due to the concept of geometric self-similarity. Thus, for a homogeneous material, such as the selected materials, the hardness is expected to stay constant with increasing indentation depth. On the other hand, the accuracy of the Oliver and Pharr method of analysis also depends on the extension of the elastic unloading under tip contact to the material surface.

The pile-up phenomenon has been extensively studied in the last two to three decades, and several works have been published so far [17,18,19,20,21]. Actually, the Oliver and Pharr method does not include the pile-up phenomenon for the quantitative evaluation of both hardness and elastic modulus. This, in turn, can generate a certain amount of overestimation of *H* and underestimation of *E_r_* measurements [20]. In this respect, Oliver and Pharr, in their review paper on nanoindentation [22], state that pile-up evaluation has to be considered still unresolved.

The main factors responsible for piling up are the ratio of the yield stress to the Young’s modulus (Y/E), and the material work-hardening behavior [23,24]. In particular, very small Y/E tends to induce pile-up phenomena; correspondingly, low degrees of work hardening generally induced pile-up in ductile materials. It was shown that Oliver and Pharr’s approach is correct for Y/E > 0.05. This low ratio is justified based on Sneddon’s analysis of surface profiles for elastic contacts, to which the Oliver and Pharr method is related [23,24,25]. For most ductile metals and alloys with Y/E in a range of 10^−4^ to 10^−2^, the Oliver and Pharr analysis can often be inaccurate, and pile-up phenomena can typically account for 5–20% inaccuracy [25]. As for the material work hardening, when this is low, the sample surface tends to flow upward over the faces of the indenter.

Comprehensive theoretical, computational, and empirical approaches have been proposed to elucidate the contact mechanics and deformation mechanisms occurring on hardness and Young’s modulus evaluation when pile-up phenomena occur (e.g., [18,26,27,28]).

The influence of pile-up and the correlation to the material work hardening and ductility behavior were examined by Bolshakov et al. using a finite element approach [29]. In their work, the authors described the material pile-up tendency by considering the ratio between the final indentation depth, *h_f_*, defined as the depth of the print after unloading, and the indentation depth recorded at peak load, *h_max_*, which, for the Berkovich indenter, lies in the range 0 < *h_f_/h_max_* < 1. The lower limit corresponds to fully elastic deformation, whereas the upper limit corresponds to a rigid-plastic behavior. Thus, the amount of pile-up becomes significant only when *h_f_/h_max_* is close to 1, whilst when *h_f_/h_max_* < 0.7, very little pile-up occurs.

On this ground, it is possible to consider the pile-up phenomenon as an indirect method for measuring the ductility of metallic materials. It is this capability that was here taken into consideration by quantifying the amount of pile-up. A ductile DHP copper in two different metallurgical statuses was used: pure Cu that was annealed (4 h at 600 °C, followed by air-cooling), and a DHP Cu cold-rolled to 70% thickness reduction (H58). Nanoindentation imprints were produced on sample surfaces polished to optical flatness.

## 3. Results and Discussion

### 3.1. Calibration Protocol

It was found that the use of the proposed functional loads for the two-step calibration protocol based on the three indenters and three reference materials (Section 2.1) greatly improved the accuracy and reproducibility of data acquisition in comparison to the standard triangular load function. Hence, the accuracy and reproducibility of both hardness and Young’s modulus values by the Oliver–Pharr approach can be greatly improved by using the proposed calibration protocol. The described calibration procedure, validated for B, cc, and Sp indenters, was previously published by Cabibbo et al. in [30], based on a round robin experiments carried out in various laboratories in Europe and Israel.

### 3.2. Indentation H and E_r_ Measurements

Figure 3 shows the microstructure of the bulk metallic materials used in the present work.

Commercially pure aluminum is characterized by a homogeneous and stress-relieved structure with equiaxial grains of 300 ± 20 μm (Figure 3a). DHP Cu shows equiaxed grains (and twins inside them) of 85 ± 8 μm mean grain size (Figure 3b). The structure of the ball bearing 100 Cr6 steel exhibits globular equiaxed ferrite of average size of 20 ± 3 μm and fine pearlite globules, together with sub-micrometer carbides (Figure 3c). The grey cast iron has a dual-phase microstructure formed by large pearlite areas within the ferrite phase, which is in the form of equiaxed grains of average size 28 ± 3 μm (Figure 3d). 

Load-displacement and H measurements were compared to those of the reference material (Fq), following the calibration protocol. Representative (out of a series of 8 × 8 indents) load-displacement curves for the different materials are shown in Figure 4.

Figure 5 shows a direct comparison of the five materials (including the reference Fq) for the same peak load (9000 μN). The load-displacement curves reported in Figure 5, by decreasing the peak load down to 1000 µN, show similar loading and unloading slopes. In particular, having the same unloading slopes for the different materials means that the reduced elastic modulus (*E_r_*), that is, local elastic modulus, complies with the material elastic modulus irrespective of the applied peak load. This is an important aspect to obtain reliable and sound indentation results that can directly be linked to the mechanical properties of the materials at a macroscopic scale.

It is noteworthy to highlight the quite similar nanoindentation response of 100 Cr6 steel and spheroidal grey cast iron (C.I.).

The raw data analysis carried out for the hardness measurement was repeated, applying an increased load and, thus, penetration depth. Figure 6 shows a well-defined plateau of the hardness with increasing contact penetration depth. The decreasing to a minimum hardness plateau is recognized as the indentation size effect (ISE) [31,32,33,34,35,36]. As expected, the ISE was not observed for Fq, since the hardness of this material is expected to be insensitive to the penetration depth [37]. The final hardness values, shown in Figure 6, are thus based on the limiting plateau value of each material.

Table 1 shows the results of nano-hardness in comparison to the measured microhardness (GPa unit) as a means to validate the nanoindentation approach. The good agreement proves the feasibility of the design calibration and used method.

The advantage of using nanoindentation over microhardness tests is the unique opportunity to obtain hardness information from nanoscale features, such as twined/untwined grains in DHP Cu, carbides/matrix in 100 Cr6 steel, and pearlite globules/ferrite grains in grey cast iron. Inherent indentation properties and mean values of the two constituent contributions are reported in Table 2 for DHP Cu, 100 Cr6, and grey cast iron.

The results listed in Table 2 show quite a good agreement with the hardness results of Table 1. On the other hand, the possibility of correlating the hardness of individual features to the straightforward hardness is clearly shown. However, the smallest microstructure features from which the hardness can be effectively measured are strictly bound to the minimum imprint size and depth being as small as ~100 nm in width, and the penetration depth being as shallow as 60–70 nm, respectively. Hence, coarse precipitates, twins, non-metallic inclusions (oxides, silicates, and the like), pearlite colonies, bainite regions, martensite ledges, and carbides in steels can be effectively sensed by nanoindentation.

### 3.3. Pile-Up Phenomenon

The pile-up phenomenon was evaluated in the annealed DHP Cu. Figure 7 depicts the model of pile-up upon nanoindentation with Berkovich imprint as a triangular projection at one of the three pile-up lobes of an indent located in-between its edge and its outer periphery. The radial distance ai corresponds to the horizontal projection of the highest ridge with respect to the undeformed material surface, where i indicates the i-th generic lobe. The true tip contact area A_true_ can be obtained by the sum of the contact area as calculated through the Oliver and Pharr analysis, A_OP_, and the contribution yielded by the entire pile-up perimeter imprint, A_PU_. The ridge of a Berkovich indent can be described by a semi-ellipse of b and a as the major and minor axis lengths (Figure 7).

The three terms ai are directly measured taking the produced indent profile image. That is, by measuring the horizontal distance from the pile-up contact point tip, T, from the edge, E, of the imprint.

The corresponding area is thus given by A_eq_ = (b^2^/4)·tan(60°) = 0.433·b^2^.

The projected contact area, A_c_, is obtained by considering the corresponding contact depth, h_c_, which traces an equilateral triangle of side b. In the present case, for a perfect Berkovich tip, Ac = 24.56·h_c_^2^ = 0.433·b^2^, from which b = 7.531·h_c_. The area of each semi-elliptical pile-up contact region is Ac = (π/4)a_i_. This, in turn, means that the total pile-up contact area is A_PU_ = πb/4∑a_i_. It is found that A_true_ = A_OP_ + 5.915·h_c_∑a_i_. This scheme is hereafter denoted as method-1. 

As proposed by Beegan et al. in [15], Saha and Nix in [17], and Zhou et al. in [24], an alternate approximation to the pile-up contact point can be that which assumes an arc around one edge of the indent and is pointed to the opposite indent corner (circle arc in Figure 7).

By using this approach, A_PU_ can be written as A_PU_ = 3[(πR^2^/6) − (√3b^2^/4)] ≅ [(2π − 3√3)/4]b^2^, where R is the pile-up arc, and b is the indent edge. Here, A_OP_ is geometrically determined as the actual area of the print as A_OP_ = (√3/4)b^2^, which, for the Berkovich tip geometry, can be written as A_OP_ = 3√3h_c_^2^tan^2^(65.3°), giving b = 2√3tan(65.3°)h_c_. A_true_ is thus given by the linear combination of A_PU_ and A_OP_. This latter relationship implies that A_true_ can be obtained indirectly from the contact depth, h_c_, without the need to physically measure the indent edge, b. This approach is here defined as method-2.

Figure 8 shows a typical scanning probe image (SPI) of pile-up in the annealed DHP Cu. The two models just described were applied to this case study, and the corresponding true area was calculated and used for correcting the hardness given by the Oliver and Pharr method (Table 3).

The values reported in Table 3 can be used to quantitatively evaluate the amount of pile-up generated by the imprint. The two methods described equally account for the ductility of DHP Cu. The amount of pile-up linearly increases with ductility. This can be inferred by the linear pile-up increment from the cold-rolled to the annealing state. This essentially geometrical feature induced by the interaction of the Berkovich indenter on the tested material is in fact a mechanical response of the material. Method-1 or method-2 can be used to estimate the material ductility through the corresponding deviation of the hardness value with respect to the value obtained by the Oliver and Pharr method without pile-up phenomenon.

To better examine this aspect, the analysis was repeated in cold-rolled DHP Cu (H58 metallurgical state).

Figure 9 showing a representative SPI of the triangular imprint, displaying the three pile-up lobes for the H58-DHP Cu. By applying the same models and equations previously used for the annealed DHP Cu, the hardness deviations to the Oliver and Pharr data were evaluated. The results are listed in Table 4.

Correspondingly, the ductility reduction induced by the cold-rolling H58 process with respect to the initial value of the annealed DHP Cu was measured by tensile test, and the resulting mean tensile-strain curves are reported in Figure 10.

The H58 cold-rolling process reduced the DHP Cu plastic strain behavior by 35%, from ε = 62.5 to 42%. At the same time, the two methods of pile-up calculation by nanoindentation showed a hardness deviation reduction from the annealed to cold-rolled state of 31–35%. That is, a straightforward relationship can be found between the nanoindentation pile-up imprint, and the corresponding ductility variation induced by the different metallurgical states of a given metallic material. Nanoindentation represents a unique and useful tool to infer the metallic material ductility behavior, in addition to measuring the hardness of characteristic sub-micrometer microstructure features and the local elastic modulus (reduced Young’s modulus).

## 4. Concluding Remarks

A calibration protocol was introduced, and this was reported to be an optimization procedure for a sound and fully reproducible hardness and reduced elastic modulus evaluation by nanoindentation.

The primary aim of the present work was to show the possibility of evaluating the hardness of sub-micrometer features in metallic materials such as pure metals (Al and DHP Cu), ferrous materials, ball bearing 100 Cr6 steel, and dual-phase grey cast iron (ferritic-pearlitic structure). It was shown that nanoindentation was able to determine the hardness contribution yield by small microstructure features such as twinning in DHP Cu, sub-micrometer carbides in 100 Cr6, and pearlite colonies in grey cast iron.

A possible means of using a local plastic deformation phenomenon occurring and induced by nanoindentation in ductile metallic material, generally recognized as the pile-up phenomenon, was also shown. This was quantitatively modelled through two similar geometrical methods. Both methods were successfully applied to DHP Cu in two metallurgical states, one annealed and the other cold-rolled. The two methods accounted for the hardness measurement deviation induced by the pile-up occurrence all around the triangular edges of the Berkovich tip print with respect to the standard Oliver and Pharr measurement. It was shown that these methods were able to exhaustively determine the amount of ductility loss from annealed to cold-rolled DHP Cu. This approach was validated by tensile stress measurements on both annealed and cold-rolled DHP Cu.

## Figures and Tables

**Figure 1 materials-14-06360-f001:**
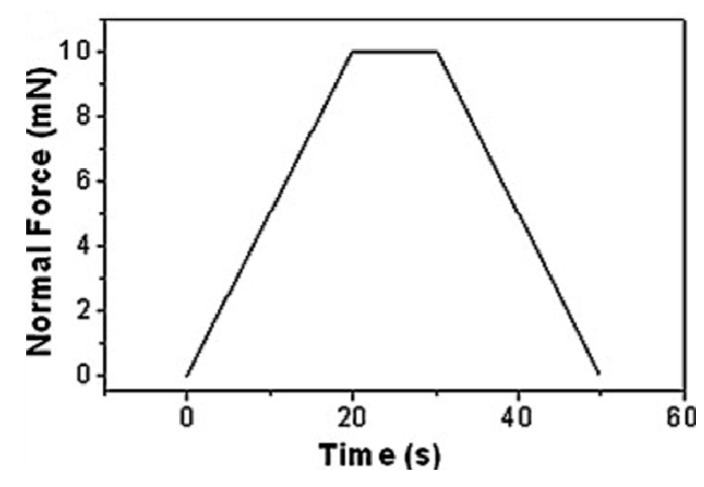
Set-up of the first step of the calibration protocol used on Fq, Sa, and Pc, consisting of a trapezoidal load functions with peak dwelling loads ranging from 0.1 to 10 mN.

**Figure 2 materials-14-06360-f002:**
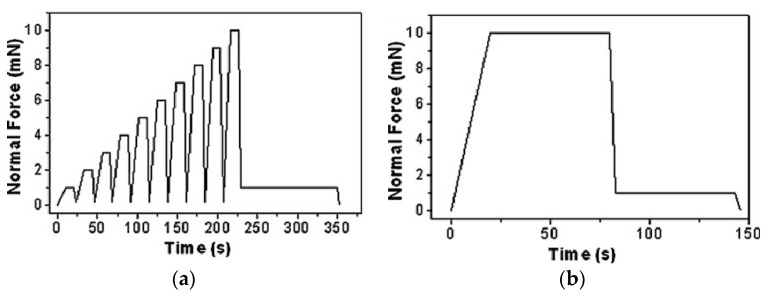
Set-up of the second step of the calibration protocol used on Fq and Sa (**a**), and Pc (**b**). The normal force of the *y*-axis corresponds to applied load.

**Figure 3 materials-14-06360-f003:**
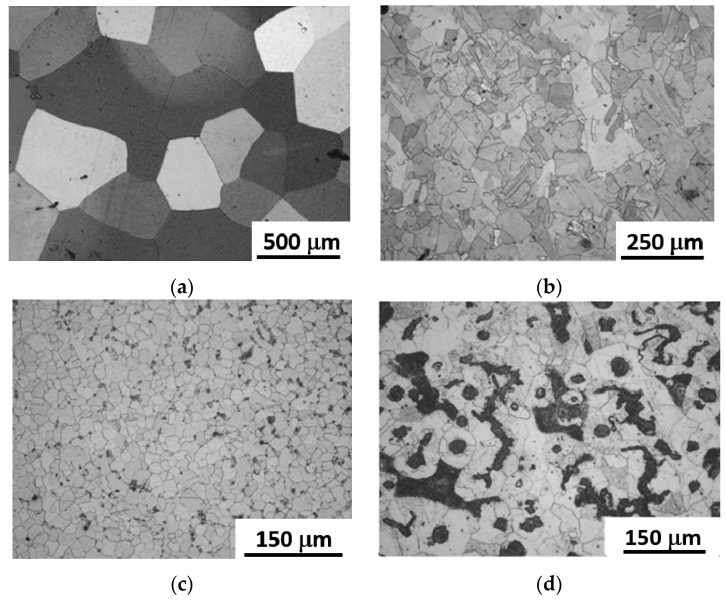
Optical micrography of commercially pure aluminum (**a**), DHP copper (**b**), 100 Cr6 steel (**c**), and cast iron (**d**).

**Figure 4 materials-14-06360-f004:**
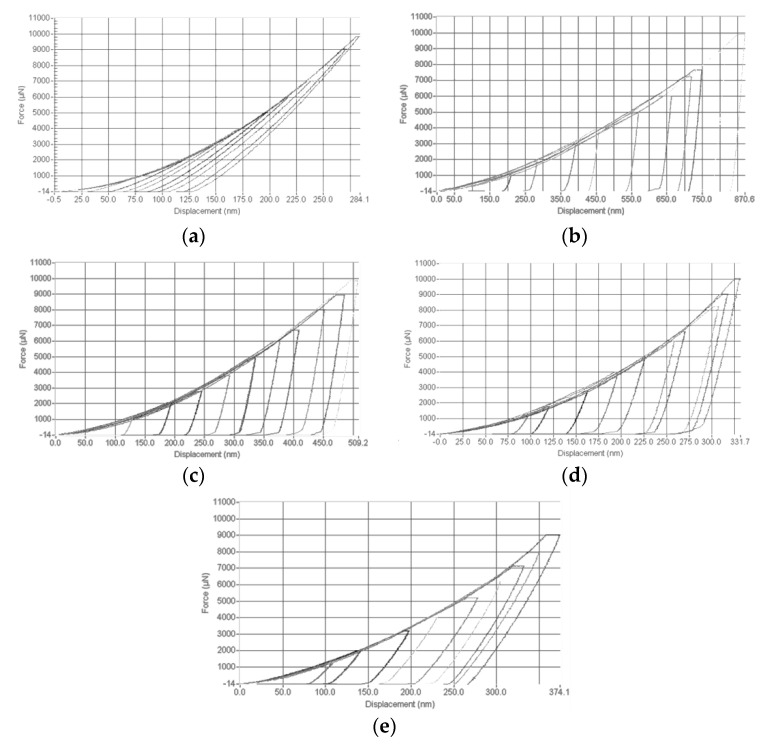
Force-displacement curves, where force is the applied load obtained for fused quartz (**a**), commercially pure aluminum (**b**), DHP copper (**c**), annealed 100 Cr6 steel (**d**), and grey cast iron (**e**). Each load-displacement curve is representative of an 8 × 8 series of individual tests.

**Figure 5 materials-14-06360-f005:**
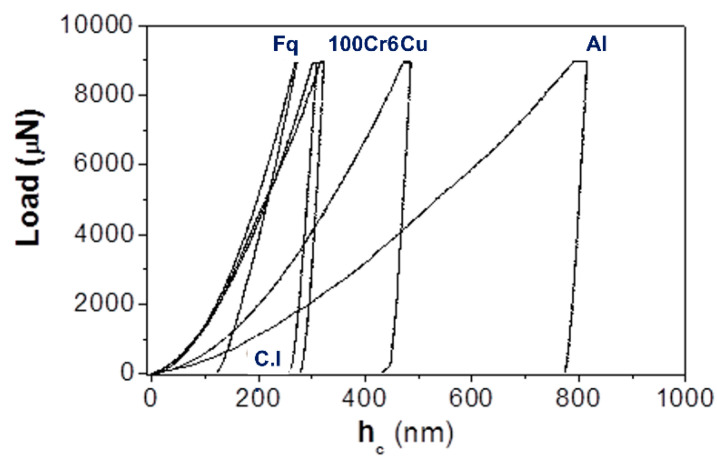
Load-displacement curves (P_Max_ = 9000 μN) for Fq, pure Al, DHP Cu, 100 Cr6 steel, grey cast iron.

**Figure 6 materials-14-06360-f006:**
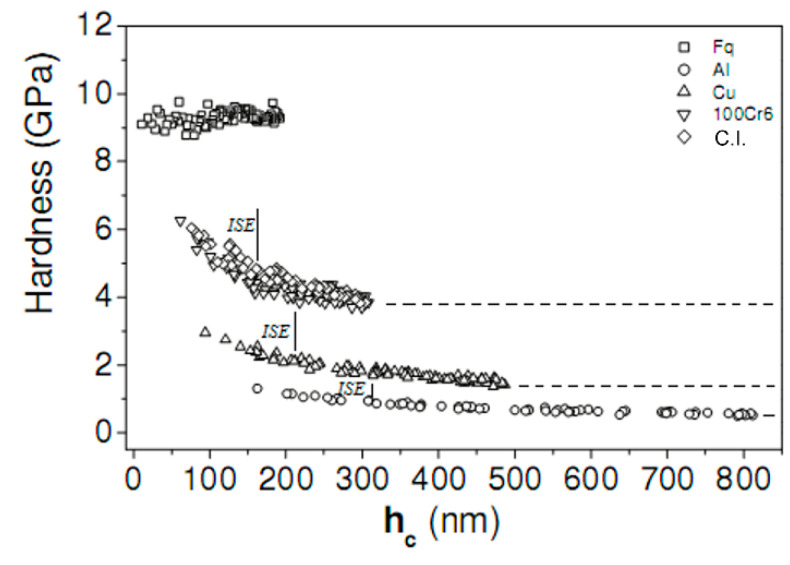
Influence of the penetration depth, h_c_, on the hardness of the five different materials. The threshold value beyond which the ISE no longer has effect is shown in the figure; as expected, this did not hold for the reference Fq.

**Figure 7 materials-14-06360-f007:**
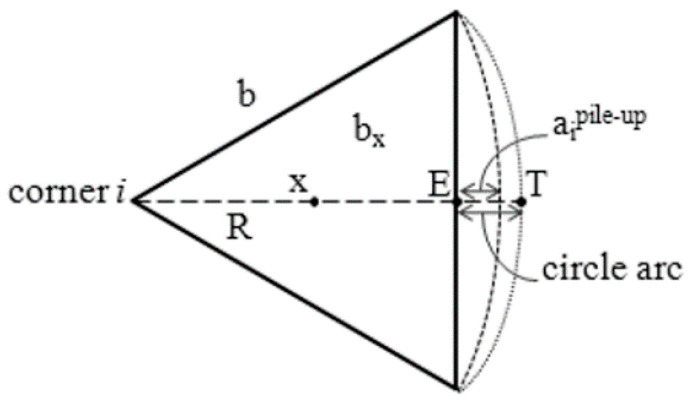
Schematic of the pile-up model. R is the arc radius (used in the method-2 model of pile-up); point T is the farthermost point from corner i of pile-up used for the evaluation of the circle arc (method-2); x is the height of the triangular projected area meeting the triangular edge at point E; a_i_^pile up^ is the horizontal projection of the tallest ridge (used in method-1 of pile-up).

**Figure 8 materials-14-06360-f008:**
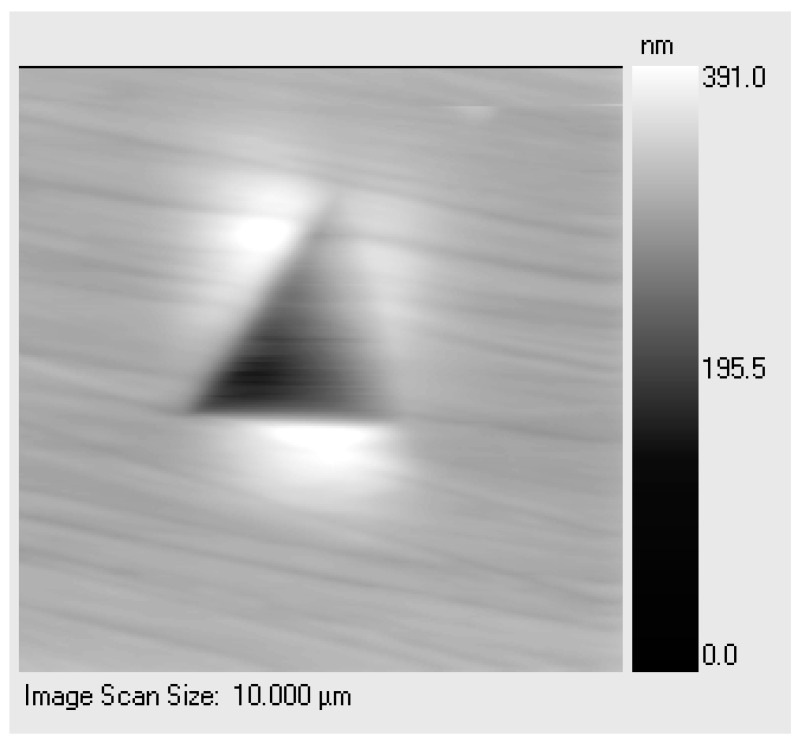
SPI of the pile-up phenomenon in annealed DHP Cu.

**Figure 9 materials-14-06360-f009:**
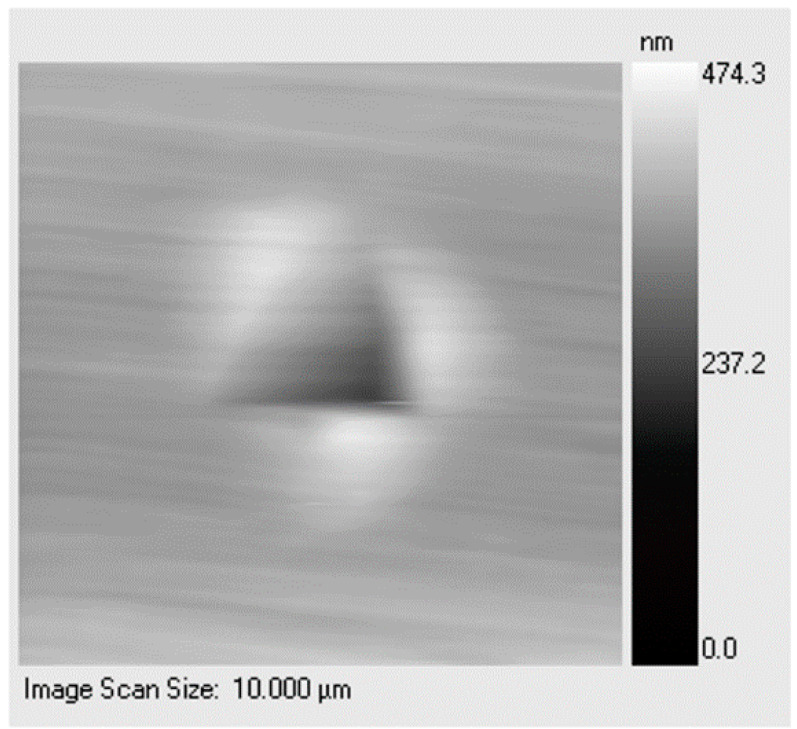
SPI of a pile-up phenomenon that occurred in the H58-DHP Cu.

**Figure 10 materials-14-06360-f010:**
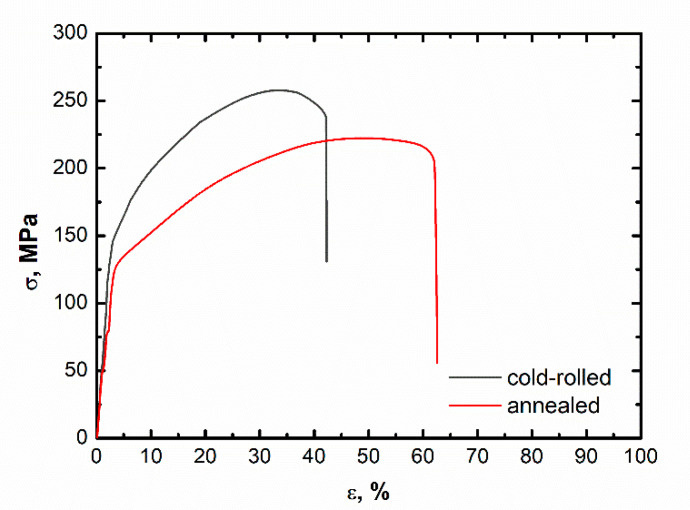
Stress-strain curves of annealed and H58 DHP Cu.

**Table 1 materials-14-06360-t001:** Average vales of hardness and microhardness.

Sample	Nanoindentation Hardness, GPa	Microhardness, GPa/HV
pure Al	0.21 ± 0.02	0.198 ± 0.02/20.5 ± 0.5
DHP Cu	0.67 ± 0.05	0.64 ± 0.05/65 ± 2
100 Cr6	1.91 ± 0.04	1.94 ± 0.08/195 ± 5
C.I.	2.45 ± 0.05	2.43 ± 0.05/230 ± 5

**Table 2 materials-14-06360-t002:** Nanoindentation hardness at the different microstructure features of DHP Cu, 100 Cr6 steel, and grey cast iron (C.I.). Constituent a/constituent b are twined/untwined grains in DHP Cu, carbides/matrix in 100 Cr6, pearlite/ferrite in grey cast iron. Mean values are calculated as geometric average.

Sample	Constituent a/Constituent b, GPa	Fraction of Constituent a/Constituent b	Mean, GPa
DHP Cu	0.74 ± 0.04/0.51 ± 0.02	0.58/0.42 ± 0.02	0.64 ± 0.08
100 Cr6	8.55 ± 0.15/1.16 ± 0.10	0.09/0.91 ± 0.02	1.83 ± 0.15
C.I.	4.30 ± 0.20/1.70 ± 0.10	0.28/0.72 ± 0.04	2.45 ± 0.10

**Table 3 materials-14-06360-t003:** Deviation to the Oliver and Pharr analysis pile-up. This was evaluated by applying method-1 and method-2.

	Hardness/% Deviation from Oliver and Pharr Value, GPa	Oliver and Pharr Value (No Pile-Up Is Taken into Account), GPa
Method-1	062 ± 0.02/−16	0.74 ± 0.02
Method-2	0.59 ± 0.02/−20	

**Table 4 materials-14-06360-t004:** Deviation of the Oliver and Pharr analysis generated by pile-up. This was calculated by applying method-1 and method-2 to the H58-DHP Cu.

Sample	Hardness/% Deviation of Oliver and Pharr Value, GPa	Oliver and Pharr Value (No Pile-Up Is Taken into Account), GPa
Method-1	0.74 ± 0.02/−11	0.83 ± 0.02
Method-2	0.72 ± 0.02/−13	

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
