# Peer review of "Instrumented Nanoindentation Tests Applied to Bulk Metallic Materials: From Calibration Issue to Pile-Up Phenomena"

_materials, 2021, doi:10.3390/ma14216360_

Round 1
Reviewer 1 Report
Excellent paper. Some editing of English language / spell check required (f.e. author keep using "such" instead of "such as", or "chemical attack" instead of "etching").
Author Response
Thank you for your appreciation. English check was carried out throughout the manuscript text and the two mistaken terminology and use of adjective forms were corrected.
Reviewer 2 Report
In this manuscript, the instrumented nanoindentation tests have been performed for several different materials. Some new methods for calibration and for considering the pile up phenomenon have been developed. The manuscript was well written and the results are good enough for publishing. However, before the final acceptance, some issues listed below are suggested to be addressed.
- In Figure 10, the stress-strain curves of two materials were displayed. The curves show a large elastic strain limit of >5%, which is obviously incorrect because the value should be less than 2% for Cu. The error may be caused by the incorrect strain measurement during tension tests (for example, the extensometer was not used). In this case, it is not correct to describe the plastic strain.
- In line 312-313, the authors said: "the amount of pile up linearly increases with the material ductility". However, this statement was not supported by experimental evidences. Especially, why does it increase linearly?
- The symbols in Fig. 7 should be clearly described.
- The words in Figures should be clear, e.g., Fig. 1, Fig. 4
Reviewer 3 Report
The author reported an accurate and sound calibration protocol for nanoindentation tests applied to bulk metallic materials and shown a possible way of using a local plastic deformation phenomenon occurring and induced by nanoindentation in ductile metallic material. This work is quite suitable for the special issue. These results could some useful information. This manuscript could be considered if the following concerns are addressed prior to publication:
I didn’t get clear about 2.1 subsection, calibration set-up and protocols. What is the improvement of the protocols? Why does the author set the function loads of the second step like Figure 2? What is the setup based on?
All the figures are at poor resolution in peer-review vision. Labels for curves are missing in figure 4. The figures and tables are titled casually which did not deliver the description of results clearly.
Reviewer 4 Report
In this submission, the author presented an investigation with nanoindentation tests applied to bulk metallic materials and its calibration issue to pile up phenomena. In general, this manuscript has been organized well, and some interesting results have been obtained.
**Please see the attached file for more comments embedded in the pdf file along with the highlighted text.

Author Response
Comments and Suggestions for Authors
In this submission, the author presented an investigation with nanoindentation tests applied to bulk metallic materials and its calibration issue to pile up phenomena. In general, this manuscript has been organized well, and some interesting results have been obtained.
**Please see the attached file for more comments embedded in the pdf file along with the highlighted text.
Reply.
All the remarks and comments attached to the pdf file were addressed in the revised version of the manuscript. The pdf file produced by the Reviewer was updated with response to all the comments and suggestions appeared in the file.
Please see the attachment.
